# HIV-1 Impairment via UBE3A and HIV-1 Nef Interactions Utilizing the Ubiquitin Proteasome System

**DOI:** 10.3390/v11121098

**Published:** 2019-11-27

**Authors:** Dohun Pyeon, Vivian K. Rojas, Lenore Price, Seongcheol Kim, Meharvan Singh, In-Woo Park

**Affiliations:** 1Departments of Microbiology and Molecular Genetics, Michigan State University, East Lansing, MI 48824, USA; dpyeon@msu.edu; 2Department of Microbiology, Immunology and Genetics, University of North Texas Health Science Center, Fort Worth, TX 76107, USA; viviankrojas@gmail.com (V.K.R.); lenore.price@unthsc.edu (L.P.); 3Pharmacology and Neuroscience, University of North Texas Health Science Center, Fort Worth, TX 76107, USAmsingh@luc.edu (M.S.)

**Keywords:** HIV-1 Nef, UBE3A (E6AP), ubiquitin, ubiquitin proteasome system, proteasomal degradation

## Abstract

Molecular basis of HIV-1 life cycle regulation has thus far focused on viral gene stage-specificity, despite the quintessence of post-function protein elimination processes in the virus life cycle and consequent pathogenesis. Our studies demonstrated that a key pathogenic HIV-1 viral protein, Nef, interacted with ubiquitin (Ub)-protein ligase E3A (UBE3A/E6AP), suggesting that interaction between Nef and UBE3A is integral to regulation of viral and cellular protein decay and thereby the competing HIV-1 and host cell survivals. In fact, Nef and UBE3A degraded reciprocally, and UBE3A-mediated degradation of Nef was significantly more potent than Nef-triggered degradation of UBE3A. Further, UBE3A degraded not only Nef but also HIV-1 structural proteins, Gag, thus significantly inhibiting HIV-1 replication in Jurkat T cells only in the presence of Nef, indicating that interaction between Nef and UBE3Awas pivotal for UBE3A-mediated degradation of the viral proteins. Mechanistic study showed that Nef and UBE3A were specific and antagonistic to each other in regulating proteasome activity and ubiquitination of cellular proteins in general, wherein specific domains of Nef overlapping with the long terminal repeat (LTR) were essential for the observed actions. Further, Nef itself reduced the level of intracellular Gag by degrading a cardinal transcription regulator, Tat, demonstrating a broad role for Nef in the regulation of the HIV-1 life cycle. Taken together, these data demonstrated that the Nef and UBE3A complex plays a crucial role in coordinating viral protein degradation and hence HIV-1 replication, providing insights as to the nature of pathobiologic and defense strategies of HIV-1 and HIV-infected host cells.

## 1. Introduction

To date, smooth transitioning from the early to late stage of HIV-1 life cycle has been mainly explained by a stage-specific expression of viral genes in the infected cells. To elaborate, tat, rev, and nef genes are expressed at the early stage of HIV-1 infection, and the expressed Rev pumps out the structural genes harboring Rev-responsive element (RRE), such as gag, pol, and env, into the cytoplasm to produce the structural proteins at the late stage of virus life cycle in the infected cells [1,2,3,4,5]. We are also mindful, from the ensuing, that the elimination of synthesized viral proteins in a timely manner after completion of their duties could be equally critical for regulation of the phase-specific transitioning of HIV-1 life cycle and hence HIV-1-associated pathogenicity. However, the molecular details as to the elimination of synthesized intracellular viral proteins have not been comprehensively studied.

Several previous reports indicate that HIV-1 and HIV-1-infected cells exploit cellular machineries for viral and cellular protein decay at every step of the HIV-1 life cycle: Degradation of CD4 by Nef/Env/Vpu is crucial to virus entry and superinfection [6,7,8,9,10,11,12,13], degradation of Gag by TRIM5α is integral to species tropism [14,15,16,17,18,19], Vif/APOBEG (apolipoprotein B mRNA editing enzyme, catalytic polypeptide-like) determines susceptibility of HIV-1 [20,21,22,23,24,25], and degradation of Tetherin by Vpu is essential for release of the assembled virus particles [26,27,28,29], etc. However, little is known about how these individual events taking place at different stages of the virus life cycle are orchestrated and what viral and/or cellular elements are responsible for such regulation of intracellular viral and cellular protein degradation.

We hypothesize that HIV-1 Nef is the most plausible candidate to execute the molecular process of inter-regulation of the stabilities of viral and cellular proteins for the following reasons: First, unlike other HIV-1 viral proteins which exist only at a specific stage of HIV-1 infection, Nef expressed as an early gene in HIV-1 infection remains through the late stage in the host cells to be packaged into virions, as Nef is a virion protein [30,31,32]. That is, Nef has the temporal advantage of encompassing all stages for regulation of proteasomal degradation processes in the infected cells. Second, each of the other viral proteins localizes to a specific subcellular compartment. However, Nef is basically ubiquitous throughout the cytoplasm at sites including the cytoplasmic membrane [33,34,35], endosomes [30,36], ER [32,37], etc., where protein degradation occurs actively. Thus, Nef is spatially advantaged. Nef is also known to degrade a key accelerator, Tat, for HIV-1 replication via the ubiquitin proteasome system (UPS) [38], indicating that Nef even impairs global regulation of viral gene expression mediated by Tat through the proteasomal degradation pathway. Further, Nef itself is mono-ubiquitinated [39] and excludes SERING5 for the virus infection [8,40] and UbcH7, the E2 ubiquitin-conjugating enzyme, from the Nef-mediated lipid raft to inhibit c-Cbl, a E3 ubiquitin ligase [7], suggesting that Nef plays an integral role in regulating the stabilities of viral and cellular proteins, which is critical for competition between HIV-1 and the infected host cells.

To investigate whether Nef is indeed involved in regulation of intracellular viral and cellular protein decay, we examined whether Nef interacts with any critical cellular proteins for the proteasomal degradation and thus identified an ubiquitin-protein ligase E3A (UBE3A, E6AP). Since UBE3A is one of the E3 ubiquitin ligases which determine selection of target protein with specificity for degradation by ubiquitination through HECT domain for the proteasomal degradation [41,42,43,44], it is suggested that binding of Nef with UBE3A plays a critical role in regulating decay of viral and cellular proteins by the UPS in HIV-1-infected cells. Therefore, we investigated the impact of these interactions on regulation of degradation of viral and cellular proteins with respect to virus replication, to bring insight into competing survival strategies of HIV-1 and the infected cells, and foster development of anti-viral agents for physical knockout, by the UPS, of key viral pathogenic element, Nef, employing UBE3A.

## 2. Materials and Methods

### 2.1. Cells and Reagents.

Jurkat T and 293T cells were cultured in RPMI1640 and Dulbecco modified Eagle medium (DMEM), respectively, supplemented with 10% heat-inactivated fetal bovine serum (FBS) and 1% penicillin/streptomycin. Employed antibodies (Ab) and reagents are: Anti-Myc (9E10), −UBE3A (H-182), and −HA (F-7) antibodies were purchased from Santa Cruz (Santa Cruz, CA, USA), anti-Flag and -β-actin (O61M4808) antibodies were obtained from ThermoFisher Scientific (Waltham, MA, USA) and Sigma (St. Louis, MO, USA), respectively, and Alexa Fluor 555 conjugated to a secondary antibodies were purchased from Life technologies (Grand Island, NY, USA). Other reagents utilized for these experiments are Cycloheximide (Sigma), MG132 (Cayman, Ann Arbor, MI, USA) and siRNA specific for UBE3A (Sigma).

### 2.2. Plasmids.

BE3A-expressing plasmid (pUBE3A) was constructed by inserting the open reading frame of UBE3A tagged with Flag epitope in place of USP15 in the pCR-BluntII-topo-USP15 (clone ID #40118994, Open Biosystems), using NotI and SalI restriction enzymes. The coding region of nef of HXBc2 strain of HIV-1 was cloned into EcoRI and BamHI sites of pCDNA3.1(−)-Myc.His (Agilent, Santa Clara, CA, USA) to generate pNef.Myc (pNef). pNef.GFP plasmid encoding Nef.GFP fusion protein was constructed by placing nef-coding region in pNef into pEGFP-N3 (Takara.Clontech). Ubiquitin-expressing plasmid tagged with HA epitope (pUb-HA) and pZsProSensor were purchased from Addgene (Cambridge, MA, USA) and Clontech (Mountainview, CA, USA), respectively. The psPax2 was obtained through the NIH AIDS Research and Reference Reagent Program, Division of AIDS, NIAID, NIH: psPAX2 (Cat# 11348) from Didier Trono [17]. Δnef-HIV-1 was constructed by substitution of 3 (the 6th to 8th from translation initiation site of nef open reading frame) with 2 nucleotides, resulting in reading-frame shift followed by termination codon. Plasmids expressing mutant nef were generated by QuickChange II site-directed mutagenesis kit (Agilent, Santa Clara, CA, USA), using specific primers, according to the manufacturer’s protocol.

### 2.3. Yeast and Mammalian Two-Hybrid Assays

The yeast two-hybrid assay was performed according to the manufacturer’s protocol (Clontech, Palo Alto, CA, USA). Briefly, HIV-1 and SIVpbj1.9 *nef* genes in a pLexA-binding domain (BD) fusion vector (His+) and a Jurkat cDNA library expressed in a pB42-activation domain (AD) fusion vector (Trp+) were introduced into yeast strain EGY48 by co-transformation, and positive colonies were screened twice to eliminate false positives. pB42AD-cDNA plasmids were then recovered from positive colonies, sequenced and introduced into EGY48/p8op-lacZ/nef by transformation to confirm the interaction with HIV-1 and SIVpbj1.9 Nefs. Except for the cells, the mammalian two-hybrid assay was performed essentially the same as the yeast two-hybrid assay. Briefly, *nef* expressers in a pM-BD fusion vector (Clontech) and UBE3A in a pVP16AD fusion vector were introduced by co-transfection into NIH 3T3 cells with a reporter gene, pG5CAT, and pCMV-β-gal to control for transfection efficiency. Three days after transfection, chloramphenicol acetyltransferase (CAT) enzymatic activity was measured as per the manufacturer’s protocol (Clontech).

### 2.4. β-galactosidase (β-gal) Assay

Yeast strain EGY48/p8op-lacZ was co-transformed with wild-type *nef* in pLexA and with UBE3A in pB42AD. Following selection from nutrition-deficient media, transformed colonies were cultured in liquid medium until log phase, measured at 600 nm. To determine the binding affinity of Nef with UBE3A, β-gal activity in the transformed yeast was quantitated as per the manufacturer’s protocol (Clontech). The units of β-gal activity were calculated by the following equation: Miller units = (A_420_ × 1000) / (A_600_ × time_min_ × volume_mL_).

### 2.5. Transfection and Infection

Transfections of plasmid or siRNA into Jurkat T and 293T cells were achieved by Amaxa cell line Nucleofactor (Lonza, Allendale, NJ, USA), according to the manufacturer’s protocol and calcium phosphate method, respectively. Infection of HIV-1 into Jurkat T cells was performed by adding virus corresponding to 10,000 cpm reverse transcriptase (RT) activity to 1 × 10^6^ cells, and replication of HIV-1 was monitored every 3 days by measuring RT activity in the culture supernatant, as described [45].

### 2.6. Cycloheximide Determination of Protein Half-Life

To investigate whether the observed reductions in the amount of UBE3A and Nef were due to the degradation of the expressed proteins, cells transfected with pUBE3A and/or pNef were treated with 40 μg/mL of cycloheximide (CHX) (Sigma Aldrich, St. Louis, MO, USA) at 48 h post-transfection for the indicated time periods, and changes to protein levels were determined by WB analyses, as described above.

### 2.7. Immunoprecipitation (IP) and Western Blot (WB) Analysis

Cells were washed twice in ice-cold PBS, suspended in the lysis buffer containing 50 mM Tris-HCl pH 7.4, 300 mM NaCl, 1% NP-40, 50 mM NaF, 1 mM NaVO4, 1 mM PMSF and 1× protease inhibitor cocktail (Calbiochem, La Jolla, CA, USA), and incubated on ice for 20 min. After centrifugation at 20,000× *g* at 4 °C for 20 min, the supernatants were collected and saved as cell lysates. The lysates were then employed for IP and WB analyses, as described [46]. The IP and/or WB analyses in the figures are representative of multiple independent experiments.

### 2.8. Data Analysis

All values are expressed as means +/− SD of triplicate experiments. All comparisons were by a controlled two-tailed Student’s *t*-test. A *p* value of <0.05 was considered statistically significant (*), and *p* < 0.01 highly significant (**).

## 3. Results

### 3.1. Nef Interacted with UBE3A

To identify cellular proteins interacting with Nefs of HIV-1 and SIVpbj1.9, we performed the yeast two-hybrid analysis, using Jurkat cDNA library and retrieved UBE3A interacting with both Nefs. As shown in Figure 1A, our quantifiable β-gal assay showed that significant amount of β-gal activity was detected only when both, not either or neither, of Nef and UBE3A were expressed, demonstrating the specificity of the interaction between Nefs of HIV-1 or SIVpbj1.9 and UBE3A. To verify the interaction of UBE3A with the HIV-1 and SIVpbj1.9 Nefs in mammalian cells, a mammalian two-hybrid assay was performed. Co-transfection with the UBE3A expresser along ith either the HIV-1- or SIVpbj1.9-*nef* expresser resulted in strong CAT enzymatic activity, while little activity was detected in cells transfected with either *nef*- or UBE3A-expresser (Figure 1B). These data indicated that interaction of UBE3A with Nef is a common feature, taking place in both HIV-1 (HXB2 strain) and SIV. Thus, we further confirmed interactions between Nef and UBE3A by IP/WB analyses. WB analysis detected UBE3A, including the endogenous one (UBE3A band in Nef, top panel, Figure 1C), and Nef from the cells transfected with pNef.Myc and co-transfected with pUBE3A and pNef.Myc (Figure 3, the 2^nd^ panel). IP/WB showed that Nef was co-precipitated with UBE3A (Figure 1C, the 3rd panel from the top), and vice versa (Figure 1C, the last panel from the top). Further, the confocal analysis showed that both Nef and UBE3A were co-localized to the cytosol (Figure 1D), even if punctae of UBE3A were detected in the nucleus (Figure 1D). Taken together, these data confirmed physical interactions between Nef and UBE3A.

### 3.2. UBE3A Inhibited HIV-1 Replication.

Next, we investigated the significance of these interactions for the HIV-1 replication in Jurkat T cell which supports robust HIV-1 replication. Prior to executing this experiment, we first tested the efficiency of siRNA (siRNA/3A) (Sigma) in specific silencing of UBE3A expression to ascertain specificity of UBE3A action. As shown in Figure 2A, expression of the endogenous and the transfected UBE3A was inhibited, by approximately 98% (left two lanes) and 65% (right two lanes), by siRNA/3A, respectively, but not by its scrambled control siRNA/3A, indicating that the transfected siRNA was specifically reduced in expression of UBE3A. To investigate impacts of UBE3A on HIV-1 replication, we infected HIV-1 into Jurkat, and the infected cells were expanded by 3-fold dilution with the fresh RPMI1640 at every 3 days, while measuring RT activity in the culture supernatants. The infected cells were then equally aliquoted on 6 days post-infection, and the aliquoted cells were transfected with different doses of pUBE3A, of siRNA/3A, or of non-target siRNA. Impacts of UBE3A on the changes to the amount of intracellular viral protein, p24, and replicability of HIV-1 were determined on 9 days post-infection by WB and RT assay, respectively. Our data showed that UBE3A significantly reduced the amount of intracellular p24 and impaired HIV-1 replication in parallel (Figure 2B). Transfection of siRNA/3A (siR/3A) into Jurkat impaired expression of UBE3A and expunged UBE3A-triggered inhibitory effects on p24 accumulation and HIV-1 replicability (Figure 2), suggesting that the observed blockade of HIV-1 replication was UBE3A-specific. Further, UBE3A was not detected in the virions, and replication kinetics of the progeny viruses generated from pUBE3A-transfected Jurkat cells was basically identical to that from the isotype plasmid-transfected Jurkat. Taken together, these data indicated that UBE3A plays a critical role in protection of the HIV-1-infected cells by reducing the amount of the intracellular Gag and thereby suppressing replication of HIV-1 in the HIV-1-infected cells.

We then examined whether the observed reduction of p24 and inhibition of HIV-1 replication had resulted from UBE3A-mediated impairment of HIV-1 viral gene expression. To this end, we transfected pUBE3A or pNef.Myc with HIV-1 LTR promoter fused with firefly luciferase reporter gene (pLTR-FLuc) into Jurkat, and alterations to the promoter activity by expression of UBE3A or Nef were measured. Consistent with the previous report [47], Nef enhanced the promoter activity, and UBE3A also slightly augmented LTR-promoter activity (Figure 2C), suggesting that the observed inhibition of HIV-1 replication by UBE3A was not due to the impairment of viral gene expression.

### 3.3. UBE3A Required Nef for UBE3A-Mediated Inhibition of HIV-1 Replication

Next, we investigated the significance of interaction between Nef and UBE3A for UBE3A-triggered inhibition of HIV-1 replication. To this end, we transfected wt- and Δnef-HIV-1 provirus together with the identical increasing amount of pUBE3A into 293T cells which allow a single HIV-1 replication cycle and thus avoid complications in elucidation of molecular role of Nef from continuous culture. The RT assay was then performed to determine supernatant virus particle levels. The results showed that increasing expression of UBE3A gradually reduced the amount of intracellular p24 from the wt-HIV-1 replicating cells (Figure 3A). However, little or no reduction of p24 was observed from Δnef-HIV-1 replicating cells, even if the amount of UBE3A was increased (Figure 3A). In corroboration, increasing UBE3A gradually lowered the amount of virus particles in the culture supernatants determined by RT assay from wt-, but not from Δnef-HIV-1 (Figure 3A), concomitantly with the level of p24, indicating that Nef was essential for UBE3A-mediated reduction of the intracellular Gag proteins and thereby HIV-1 replication.

Next, we investigated whether Gag and Env, as another structural protein control, can degrade UBE3A or are just targets to be degraded by UBE3A. Accordingly, we transfected identical amounts of pUBE3A together with the increasing amounts (1, 2, 4 μg) of psPax2 [17] and of Env-expressing (pEnv) plasmids, and changes in the amount of UBE3A were examined by WB analysis. The data showed that the band intensity of p24 and Env increased, when raising the amount of Gag- or Env-expressing plasmids (Figure 3B). However, there was no decrease in the band intensity of UBE3A by increasing psPax2 or Env (Figure 3B), indicating that the levels of intracellular Gag and Env were regulated by UBE3A, not vice versa. 

### 3.4. The Amount of the Intracellular UBE3A and Nef was Mutually Regulated.

UBE3A-mediated inhibition of HIV-1 replication took place only in the presence of Nef; that is, the amount of intracellular UBE3A and Nef in HIV-1-infected cells determines replicability of HIV-1. The next question that we should ask then is how UBE3A and Nef work together to regulate HIV-1 replicability. As shown above in Figure 2, the amount of the endogenous UBE3A in HIV-1 replicating Jurkat was reduced (Figure 2B, pC3), compared with the amount in uninfected Jurkat cells (Figure 2B, Mock). Consistently, the level of UBE3A in wt-HIV-1-replicating cells was clearly less than that in Δnef-HIV-1-replicating cells (Figure 3A, top panel), even if the equivalent amount of pUBE3A were transfected. These data suggest that the amount of the intracellular UBE3A was regulated by Nef. Hence, we investigated the possibility of inter-regulation of these proteins. To this end, we transfected the increasing amount of pNef.Myc (lanes 1–3) and pUBE3A (lanes 4–5) and analyzed alterations to the amount of each protein by WB analysis. As shown in Fig. 4, both proteins were expressed in a dose dependent manner (Figure 4A,B), when they were expressed independently. However, the amount of Nef protein was gradually reduced, as the expression of UBE3A was increased (lanes 7–9, Figure 4A,B). Similarly, the amount of the intracellular UBE3A was decreased, as pNef.Myc was increased (lanes 10–12, Figure 4A). However, the reductions in band intensities of UBE3A in Figure 3A and Figure 4A were disparate, which could be due to the differential amount of Nef expressed from the wt-HIV-1 replication and from nef-expressing plasmid, respectively. Further, increases of UBE3A (lanes 7–9) and Nef (lanes 10–12) were not compatible with those of the identical amount of pUBE3A (lanes 4–6) and pNef.Myc (lanes 1–3, Figure 4A,B) in the presence of Nef and UBE3A, respectively. Specifically, the intensity of Nef bands in the presence of UBE3A (lanes 10–12, Figure 4) was appreciably weaker than that of Nef in the absence of UBE3A (lanes 1–3, Figure 4), even if the amount of the input pNef.Myc were equivalent, wherein UBE3A-triggered reduction of Nef was more potent than Nef-mediated diminution of UBE3A (Figure 4A,B). Identical results were obtained with the M-tropic YU2 strain of HIV-1 Nef.

We further confirmed the observed reciprocal reduction of UBE3A and Nef by their mutual impairment of promoter activity, taking advantage of UBE3A- and Nef-triggered augmentation of promoter activities, as shown above in Figure 2C. As shown in Figure 4C, enhancement of UBE3A-triggered promoter activity was impaired by the expression of Nef (compared lanes 4–6 with 7–9), and increase of pNef.Myc lowered UBE3A-mediated promoter activity (compared lanes 10–12 with lane 6, Figure 4C) in parallel with the changes in the level of proteins of Nef and UBE3A (Figure 4A). Taken together, these data showed that Nef and UBE3A were reduced reciprocally in the expressing cells in a dose dependent manner (Figure 4), and potent reduction of Nef by UBE3A suggested that UBE3A could be employed for the physical knock-out of Nef in the HIV-1-infected cells.

### 3.5. Reciprocal Reduction of Nef and UBE3A was due to Degradation of the Expressed Proteins.

Since both pNef.Myc and pUBE3A were expressed from the identical promoter of cytomegalovirus (CMV), and the total amount of the promoters for expressions of Nef, UBE3A and luciferase was adjusted with the corresponding isotype plasmid, pCDNA3 (pC3) to be equivalent in above experiments, mutual reduction of Nef and UBE3A in the presence of each other could have resulted from modulation of stabilities of the expressed protein, not expression of the introduced genes. To characterize the nature of mutual reduction of Nef and UBE3A, the cells transfected with pNef.Myc- and/or pUBE3A-expressing plasmids were treated with the translation inhibitor, cycloheximide (CHX) [18] for the indicated time periods, and turn-over rates of the amounts of the synthesized Nef and UBE3A proteins were determined by WB with BioRad image quantitation based on β-actin changes (Figure 5A). The results showed that nominal reduction of Nef and slight increase of UBE3A over the time period of treatment with CHX were observed, when Nef and UBE3A were expressed independently (Figure 5A). In contrast, decreases of the amount of UBE3A and Nef were discernible, when both genes were co-expressed (Figure 5A), indicating that mutual reduction of UBE3A and Nef was due to the decay of the synthesized proteins in the presence of each other.

We then investigated whether the reciprocal degradations of UBE3A and Nef were occurring through proteasomal degradation pathways, using MG132, a proteasomal degradation inhibitor [19,38,48]. The results showed that first, consistent with the above data, the amount of both Nef and UBE3A was decreased in the presence of UBE3A and Nef, respectively, and second, accumulation of the proteins in the presence of the proteasomal degradation inhibitor, MG132, was significantly more than that with dimethylsulfoxide (DMSO) (Figure 5B), demonstrating that the proteasomal degradation pathway played an integral role for the observed degradation of Nef and UBE3A. 

### 3.6. Nef and UBE3A were Antagonistic in Ubiquitination of Cellular Proteins and Regulation of Proteasome Complex

The role of UBE3A in the proteasomal degradation has been well-established [10,12,49,50]. However, regulation of degradation of cellular protein by Nef via proteasomal degradation pathway is not well-known, except for mono-ubiquitination of the protein [14] and exclusion of UbcH7, the E2 Ub-conjugating enzyme, from the Nef mediated lipid raft to inhibit c-Clb, a Ub ligase [51]. We thus further investigated molecular processes of the observed reciprocal regulation of the stability of Nef and UBE3A by the UPS by examining the capacity of Nef and UBE3A for ubiquitination of target proteins. In corroboration with the well elaborated UBE3A function [12,49,50,52], UBE3A enhanced ubiquitination of cellular proteins (Figure 6A, the 3rd lane vs. the 2nd lane from left), whereas interestingly, Nef impeded ubiquitination of cellular proteins (Figure 6A, the 4th vs the 2nd lanes). Moreover, UBE3A-mediated ubiquitination of cellular proteins was obviated by increasing Nef (Figure 6, the last two lanes), and the Nef-triggered inhibitory effect on ubiquitination of target proteins was gradually alleviated by increased expression of UBE3A (Figure 6A, lanes 5–7). Again, consistent with the above data, the amount of Nef was reduced with increasing UBE3A, and vice versa (Figure 6, lower panel). Taken together, these data demonstrated that Nef indeed played a role in regulation of stabilities of cellular proteins by modulating the level of ubiquitination, wherein Nef and UBE3A were antagonistic.

Since Nef impaired global ubiquitination processes of cellular proteins (Figure 6A), we then asked if Nef impacts proteasomal function, another key step of degradation processes by the UPS. Increases of Nef gradually enhanced cleavage of the sensor molecule by elevating proteasome activity, thereby significantly reducing green fluorescence at 496 nm in a dose-dependent manner, compared with the control (pC3), while UBE3A did not (Figure 6B), which was measured with Proteasome Sensor Vector (pZsProSensor-1, Clontech) [16]. The observed augmentation of the proteasome function by Nef was obliterated by the presence of MG132 (Figure 6B), again showing that Nef is a critical viral protein for regulation of stabilities of cellular proteins in the proteasomal degradation pathway. In contrast, UBE3A-associated proteasomal activity was basically unchanged, in the presence or absence of MG132 (Figure 6B). Further, in corroboration with the above data (Figure 5), our WB analysis (below the bar graph) showed that the amount of Nef and UBE3A was accumulated in the presence of MG132. (Since all bands were derived from the identical membrane, as addressed in the Figure legend, band intensities can be directly comparable.) Taken together, these data indicated that Nef clearly plays an important role in protein degradation by enhancing the proteasome activity in the UPS.

### 3.7. Di-Acidic and Di-Leucine Motifs in Nef Play an Essential Role in Regulation of Ubiquitination and Proteasome Function

Nef-mediated modulation of ubiquitination of cellular proteins and proteasome activity and thereby regulation of stability of proteins was unprecedented. Thus, we consequently investigated whether Nef is indeed critical to ubiquitination and proteasome activity by introducing mutations at well-known motifs in Nef. In brief, we generated seven Nef mutations, based on the previous report [53], at the Myristoylation (Myr), basic, acidic, SH3 binding (SH3), di-arginine (Di-R), β-COP, di-leucine (Di-L) domains, and impacts of these mutations on the ubiquitination of cellular proteins and proteasomal activities were examined, as described above. Consistent with the above data, wt-Nef dramatically impaired ubiquitination of cellular proteins by approximately 90%, while elimination of the di-arginine (Di-R) motif significantly abrogated wt-Nef-triggered inhibitory effect of ubiquitination of cellular proteins (Figure 7A). Similar to Figure 6B, wt-Nef enhanced proteasome activity, increasing cleavage of the sensor molecule, whereas mutations at Di-R and β-COP motifs obliterated wt-Nef-mediated increases of proteasome activity (Figure 7B). Band intensities of Nef varied in our repeated experiments, even if we transfected identical amounts of wt- and mutant *nef*-expressing plasmids under the same CMV promoter with the identical transfection conditions, suggesting that the disparate amount of the detected Nef protein in the WB analysis could be due to the differential extraction efficiency of the expressed mutant Nef proteins by the lysis buffer, by different localization to the different subcellular compartments, such as cytoplasmic membrane, endosome, etc., by the affinity changes of the expressed Nef by mutations to the antibody, and/or different level of accumulation of the mutant Nefs. Collectively, these data demonstrated that Nef in fact regulated the level of ubiquitination of the cellular proteins and proteasome activity and that Di-R motif in Nef is the most critical element for ubiquitination of the cellular proteins, while Di-R and β-COP domains play an essential role in regulation of the proteasome activity.

### 3.8. Nef Itself can Degrade HIV-1 Viral Proteins.

The above data indicated that UBE3A was associated with Nef (Figure 1) and inhibited HIV-1 replication only in the presence of Nef (Figure 3), indicating that UBE3A-mediated reduction of Gag was achieved by interacting with Nef. Thus, the next question we asked was whether Nef itself can regulate stability of HIV-1 viral proteins and thereby HIV-1 replication. It is very well established that Nef is dispensable or slightly positive for in vitro HIV-1 replication with *nef*-mutant HIV-1 [47,54,55]. However, impacts of excessive amounts of the intracellular Nef, which might actually take place in the HIV-1 life cycle (as dealt with in the Discussion) on HIV-1 replication, have not been elucidated. To investigate roles of this overabundance of Nef on HIV-1 replication, we co-transfected increasing amounts of *nef*-expressing plasmid (pNef.Myc) together with wt-HIV-1 proviral DNA into 293T cells which only reflect intracellular events for the HIV-1 single life cycle, thus eliminating the effect of subsequent virus replication; we then performed WB analysis. Under these conditions, the amount of Nef was obviously inordinate in Figure 8A, since the total amount of Nef in Figure 8A is the sum of the Nef expressed from pNef.Myc and from wt-HIV-1 replication. As shown in Figure 8, the amount of Nef was enhanced in a dose-dependent fashion with increasing pNef.Myc input, and the increases of Nef reduced p24 in the wt- HIV-1(Figure 8A) provirus replicating cells. These data indicated that excess amounts of Nef in wt-HIV-1 replicating cells inhibited HIV-1 replication by reducing the levels of the expressed Gag protein. 

Nef-triggered decreases of the intracellular p24 could be due to the degradation of Tat by Nef, as reported [38], yielding reduction of the viral gene expression and/or decay of p24 directly by Nef. Even if elucidation of molecular mechanisms for how the cytoplasmic Nef degraded nuclear Tat protein were beyond the scope of this study, investigating whether Nef in fact induced degradation of Tat would be crucial for illustrating Nef-mediated reduction of Gag. In corroboration with the previous report [38], the amount of Tat was indeed significantly lowered in the presence (lanes 8–10), but not in the absence of Nef (lanes 5–6, Figure 8B). However, the amount of Nef was basically unaltered, even if Tat expression were enhanced (lanes 8–10, Figure 8B), while increases in Nef gradually diminished Tat (lanes 11–13, Figure 8), suggesting that Nef degraded Tat, but not vice versa. The corresponding HIV-1 LTR-promoter activity measured by the Firefly luciferase (FLuc) activity was also in parallel with the amount of change in Tat (Figure 8B). However, the Tat levels were dramatically reduced in the presence of Nef, but decreases in the luciferase activity were not comparable to the amount of change in Tat (lanes 8–10 vs. lanes 5–7, Figure 8). This gentle reduction of luciferase activity, compared with the sharp decrease of the amount of Tat protein, could be due to the compensation by Nef-triggered enhancement of LTR promoter activity, even if not as potent as under Tat, as observed in Figure 2. Lanes 11–13 in the same Figure clearly exhibited gradual decrease in the amount of Tat and its corresponding reduction of the luciferase activity by increasing Nef, confirming the previous report that Nef degraded Tat [38] and suggesting that the observed reduction of p24 by the inordinate amount of Nef in Figure 8 could have arisen from Nef-mediated degradation of Tat and thus inhibition of viral gene expression.

Since UBE3A required Nef for degradation of Gag (Figure 3), excess Nef reduced the level of intracellular p24 (Figure 8), and Nef degraded Tat (Figure 8B), the next question was whether the observed decreases of p24 were fully due to impediment of viral gene expression by Nef-mediated degradation of Tat. To answer this question, we investigated whether Nef was involved in determination of the stability of Gag, using psPAX2 [47], which is deficient in *env*, *tat,* and *nef* genes, and thus excludes the effect of Tat-mediated viral gene expression. Our data showed that the amount of p24 was basically unchanged, even if the amount of pNef. Myc were increased (Figure 8C), indicating that Nef down-regulated the amount of intracellular Gag by decaying Tat and thereby incapacitating viral gene expression, not by direct degradation of Gag. Taken together, these data demonstrated that HIV-1 Nef regulated the amount of intracellular viral proteins by global modulation of viral gene expression via alteration of Tat protein stability and indicated that mutual regulation of the stability between Nef and UBE3A in HIV-1-infected cells plays a cardinal role in restriction/competition of proliferation of the infected HIV-1 and the invaded host cells.

## 4. Discussion

The significance of stage-specific expression of HIV-1 viral genes is well addressed in the investigations of the HIV-1 replication cycle. However, molecular regulation of elimination of the synthesized intracellular viral proteins, post-function, has not received thorough attention, even where such processes are integral to the viral life cycle and attendant virulence. Our above data clearly showed that reciprocal regulation of proteasomal degradation of Nef and UBE3A played a critical role in regulation of other viral and cellular protein decay and hence competing survivals of the proliferating HIV-1 and the infected cells, offering insights into pathobiologic and defense strategies between virus and host cell, and opening a new vista for obtaining, by the UPS, physical knockouts of vital pathogenic determinant, HIV-1 Nef, deploying UBE3A.

With respect to the viral pathogenesis, UBE3A associates with E6 protein (and thus it is also called as E6 associated protein—E6AP) of human papilloma virus (HPV), and E6/UBE3A complex targets the tumor suppressor, p53, for ubiquitination and degradation, thereby contributing to HPV-induced cervical carcinogenesis [53,56,57]. Interestingly, HIV-1 Nef binds to and induces degradation of p53, thus protecting HIV-1-infected cells against p53-mediated apoptosis, which results in enhancement of HIV-1 replication by prolonging the viability of the infected cells [58,59,60]. These reports suggest that UBE3A and Nef could also associate each other, sharing p53 as a common denominator. In fact, our data showed that UBE3A and Nef interacted with each other and mutually regulated their stabilities. Further, UBE3A degraded not only Nef but also other viral proteins, such as Gag and Env, only in the presence of Nef, thereby restricting HIV-1 replication, indicating that reciprocal regulation of the stabilities between UBE3A and Nef by the proteasomal degradation pathway plays an important role in regulation of HIV-1 replication. However, molecular mechanisms on how stabilities of these individual proteins are mutually regulated, while orchestrating other viral and cellular proteins by the UBE3A and Nef complexes, will need further examination as to the competition between HIV-1 and the infected cells, and as to effects on viral pathogenicity.

It is well founded that Nef is basically dispensable for in vitro HIV-1 replication [47,54,55]. What, then, is the significance of the reciprocal degradation between UBE3A and Nef and of UBE3A-mediated impairment of HIV-1 replication in the presence of Nef? That is, if Nef were dispensable for HIV-1 replication, UBE3A-mediated degradation of Nef would not influence HIV-1 replicability, unless UBE3A directly degraded Gag by proteasomal decay to thereby inhibit HIV-1 replication. Consistent with the previous reports [47,54,55], deletion of the nef gene did not alter the amount of intracellular p24, leaving HIV-1 replicability basically unchanged. However, excessive Nef produced by co-transfection of wt-HIV-1 and pNef.Myc was clearly essential in reducing the level of Gag and thereby HIV-1 replication (Figure 8), indicating that the non-stoichiometric overabundance of Nef in the wt-HIV-1-infected cells functions as a negative regulator for HIV-1 replication. Hence, how can we explain UBE3A-induced inhibition of wt-HIV-1 replication shown in Figure 2, wherein the amount of Nef is stoichiometric? Figure 2B clearly showed that HIV-1 replication was robust, when the amount of UBE3A was below the level of the endogenous UBE3A (Figure 2B, Mock) or expression of UBE3A was blocked by its specific siRNA (siR/3A). However, ectopic expression of UBE3A by transfection inhibited HIV-1 replication in a UBE3A dose-dependent manner (Figure 2B), suggesting that an efficient blockage of HIV-1 replication was achieved only by an excessive amount of UBE3A in the presence of the stoichiometric amount of Nef. Taken together, these data indicated that excess UBE3A and Nef were essential for the impairment of HIV-1 replication, and thus the reciprocal degradation of UBE3A and Nef stability, so as to mutually impede overabundance of these proteins in mediation of virus replication.

Are Nef levels relatively copious compared with other HIV-1 viral proteins at specific or many phases of the virus cycle in the context of replication blockade? As aforementioned, the HIV-1 life cycle has been demarcated along the lines of early and late stage viral gene expression, in that regulatory proteins, such as Tat, Rev, and Nef are expressed at the early stage of virus infection, while late regulatory and structural proteins are produced at the late stage of virus life-cycle, wherein Rev is the most critical viral element for transitioning between early to late phase by exporting messages harboring the rev-responsive element (RRE) to the cytoplasm [1,2,3,4,5] A crucial premise for a smooth transition of such virus life-cycle is that the synthesized viral proteins, after completing their duties, need to be removed on schedule by degradation processes. In distinction to other viral proteins, Nef protein expressed at the early stage of HIV-1 infection must remain in the infected cells until the protein is packaged into virions, as Nef is a virion protein [30,31,32]. Nef is known to degrade Tat by the UPS, when Tat is in excess in HIV-1-infected cells [38]. Consistently, our data indicated that Nef reduced the amount of intracellular Tat, but not vice versa (Figure 8), which implies that Nef decreases not only Tat expressed at the early stage of virus infection, but also the overall level of viral proteins, by regulating Tat stability and hence viral gene expression. Based on these data, it is reasonable to speculate that Nef protein expressed at the early stage of the virus infection remains relatively preponderant in the infected cells by removing Tat, while the excess Nef competes with UBE3A via proteasomal degradation for proliferation of HIV-1 and of the infected host cells. Further experiments are needed to clarify the mechanism of Nef-mediated proteasomal degradation of Tat, which will raise our understanding of the complex molecular actions of Nef.

Even if the role of UBE3A in the proteasomal degradation has been well-established [10,12,49,50], involvement of Nef in regulation of proteasomal degradation of viral and cellular proteins is poorly explained, though it is reported that Nef is mono-ubiquitinated [14], degrades a key transcription factor of HIV-1, Tat [38], and excludes SERINC5 for the virus infection [8,40] and UbcH7, the E2 Ub-conjugating enzyme, from the Nef mediated lipid raft to inhibit c-Clb, a Ub ligase [15]. Interestingly, our data demonstrated that Nef played a role in regulating the stabilities of cellular proteins by modulating the level of ubiquitination and by regulating the proteasome activity, wherein Nef and UBE3A were antagonistic. Further, structure and function analysis of Nef by the site-directed mutagenesis of *nef* gene demonstrated that Di-R motif in Nef was the most critical element for ubiquitination of the cellular proteins, while Di-R and β-COP domains played an essential role in regulation of the proteasome activity. For further confirmation of Nef role in regulation of ubiquitination, we executed proteomic analysis to investigate differential ubiquitination of cellular proteins in the presence or absence of HIV-1 Nef. The data showed that numerous cellular proteins were differentially ubiquitinated in the presence and absence of Nef, substantiating that Nef in fact played a pivotal role in regulation of ubiquitination of cellular proteins and thereby stability of the proteins. Taken together, these data provide concrete evidences that the viral protein Nef is crucial for regulating the stability of viral and cellular proteins.

In conclusion, our above data demonstrated that UBE3A and Nef mutually degraded viral and cellular proteins via modulation of ubiquitination and proteasome activity, and UBE3A-mediated degradation of Nef was significantly more potent than Nef-triggered degradation of UBE3A (Figure 2). Further, ectopic expression of UBE3A basically incapacitated HIV-1 replicability in the presence of Nef (Figure 2 and Figure 3), and recent publication also showed that the proteasome plays a key role in establishment of latency by preferentially inhibiting the Tat-dependent HIV-1 transcription [61]. Taken together, these data indicate that UBE3A can be employed for the physical knock out of the key determinants of HIV-1 pathogenicity via UPS-mediated protein degradation, opening a new opportunity to develop anti-HIV-1 therapeutics. Specifically, since changes in the intact UBE3A protein expression could dysregulate UBE3A-associated proteasomal degradation and thus HIV-infected cell biology, a modified UBE3A, composed strictly of the Nef binding domain and the active motif for specific degradation of viral proteins, could be employed to promote UBE3A-triggered targeted knockout of HIV-1 proteins, and to impact the virus/cell competition for survival in the host.

## Figures and Tables

**Figure 1 viruses-11-01098-f001:**
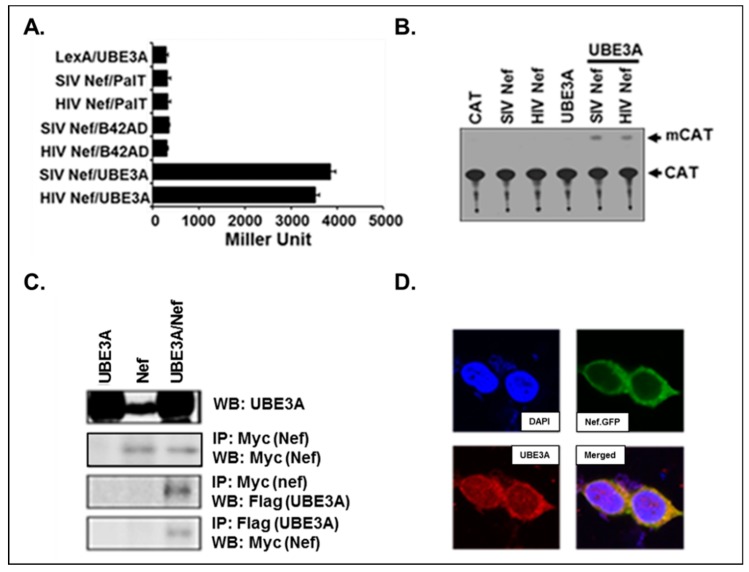
Interactions between Nef and UBE3A. (**A**) Yeast two-hybrid analysis. β-galactosidase activity was measured to evaluate the specificity of the interaction between UBE3A, HIV-1, and SIVpbj1.9 Nef using the yeast two hybrid analysis by expressing UBE3A/B42 activation domain (AD) and Nef/LexA-binding domain (LexA). Palmitoyl transferase (PalT) were used as negative controls. (**B**) Mammalian two-hybrid analysis. Chloramphenicol acetyltransferase (CAT) activity was measured, as described, and mCAT indicated modified CAT. (**C**) Immunoprecipitation (IP)/ western blot (WB). 293T cells were transfected with pNef.Myc and/or PUBE3A.Flag, and lysates from the transfected cells were immunoprocipitated, and then immunoblotted with the indicated antibodies. (**D**) Confocal analysis. 293T cells were transfected with pNef.GFP, and the cells were visualized with 4’,6-diamidino-2-phenylindole (DAPI) for the nucleus and anti-UBE3A antibody followed by Alexa Fluor 555 for UBE3A.

**Figure 2 viruses-11-01098-f002:**
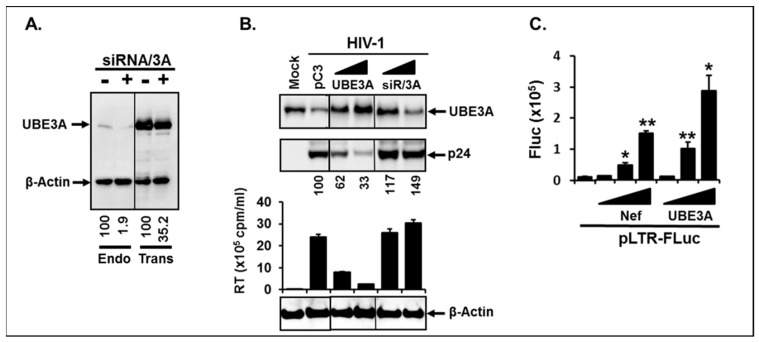
Effect of UBE3A on HIV-1 replication. (**A**) Specificity of siRNA against UBE3A (siRNA/3A). The first and last two lanes showed expression of the endogenous (Endo) and pUBE3A-transfected (Trans) UBE3A, respectively, in the absence (−) or presence (+) of siRNA specific for UBE3A (siRNA/3A). The amount of UBE3A shown as an Arabic number under the Figure was standardized with the changes of amount of β-Actin. (**B**) Impact of UBE3A on the changes of intracellular p24 and HIV-1 replication. pC3 indicated pCDNA3 to adjust total amount of the plasmid to be the same, and the Arabic numbers below the image represented the relative amount of intracellular amount of p24. The bar graph depicted the amount of virions measured by reverse transcriptase (RT) activity in the culture supernatants (1 mL). (**C**) Effect of Nef and UBE3A on HIV-1 LTR promoter activity. One microgram of plasmid expressing Firefly luciferase (FLuc) from HIV-1 LTR (pLTR-FLuc) was co-transfected with 0.5, 1, and 2 µg of pNef.Myc or pUBE3A.Flag into 293T cells, and luciferase activity from the transfected cells was measured. This luciferase assay data are representative of three independent experiments with triplicates.

**Figure 3 viruses-11-01098-f003:**
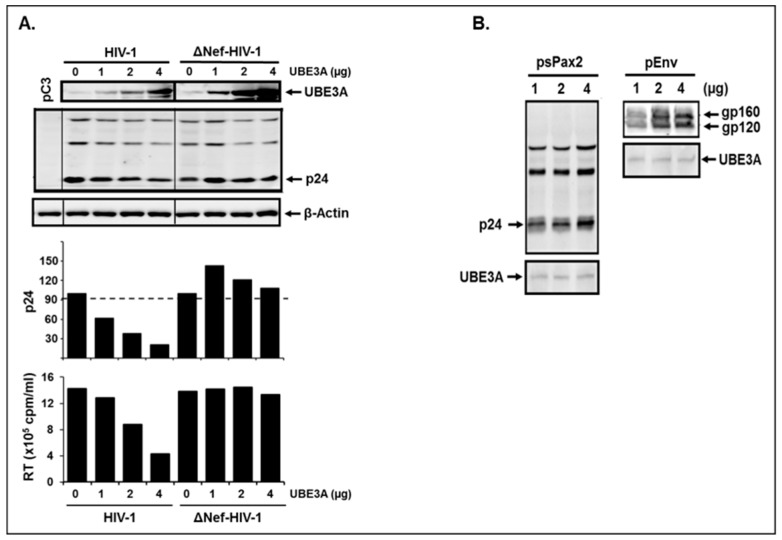
Impact of UBE3A on wt- and Δnef-HIV-1 replication. (**A**) shows changes of the amount of intracellular p24 and RT in the culture supernatants from wt- and Δnef-HIV-1-transfected cells, as increasing the amount of UBE3A and (**B**) that of UBE3A, as the amount of gag- (psPax2) and env- (pEnv) expressing plasmids was increased. The total amount of the transfected DNA was adjusted with the isotype plasmid pC3 level remaining the same.

**Figure 4 viruses-11-01098-f004:**
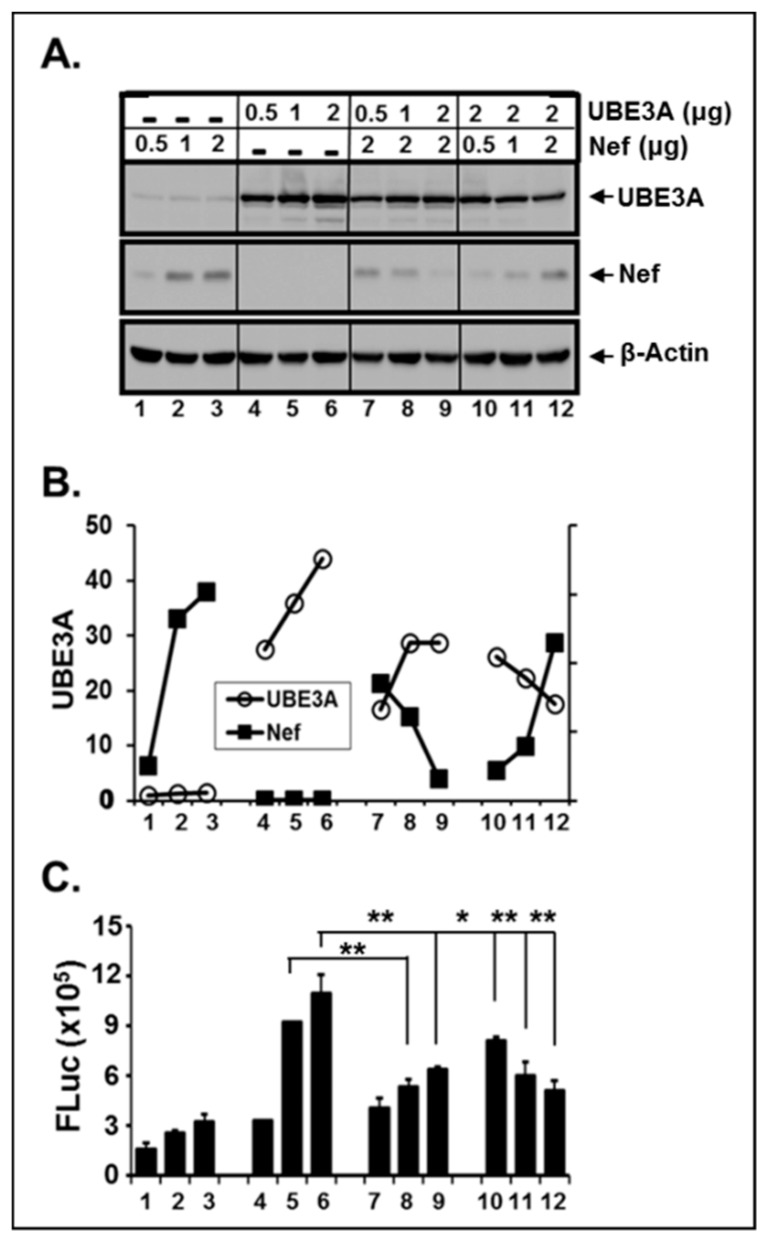
Mutual regulation of the amount of intracellular UBE3A and Nef. (**A**) Increasing amount (0.5, 1, and 2 μg) of pUBE3A.Flag or pNef.Myc alone or together with 2 ug of pNef.Myc or pUBE3A, respectively was transfected into 293T cells, and changes of the amount of UBE3A and Nef were analyzed by WB. (**B**) Line graph shows quantification of the relative intensities of UBE3A (open circle) and Nef (closed square) bands in the panel (**A**). (**C**) The bar graph represents FLuc, when the indicated amounts of pUBE3A and/or pNef.Myc were co-transfected with 1 μg pLTR-FLuc into 293T cells. This luciferase assay is representative of three independent experiments with triplicates.

**Figure 5 viruses-11-01098-f005:**
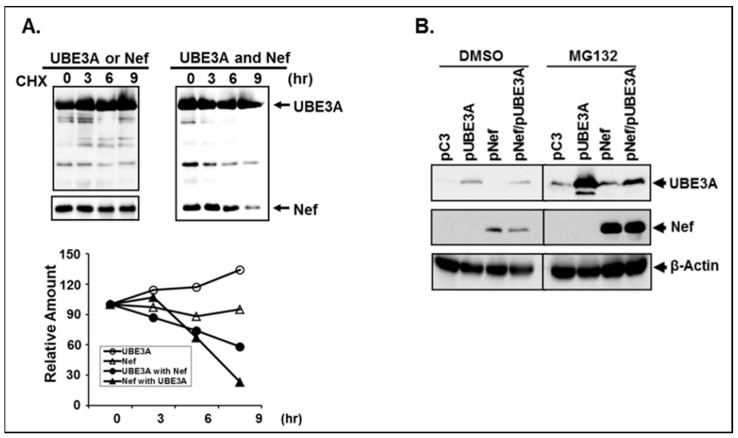
Degradation of UBE3A and Nef. (**A**) The upper shows WB analysis to determine the amount and turn-over rate of the synthesized UBE3A and Nef over the indicated time-period of the treatment of the cells with cycloheximide (CHX), and the lower line graph represents quantification of changes of the amount of UBE3A and Nef. (**B**) shows accumulation of Nef and UBE3A expressed independently and/or together in the presence of proteasomal (MG132) degradation inhibitor.

**Figure 6 viruses-11-01098-f006:**
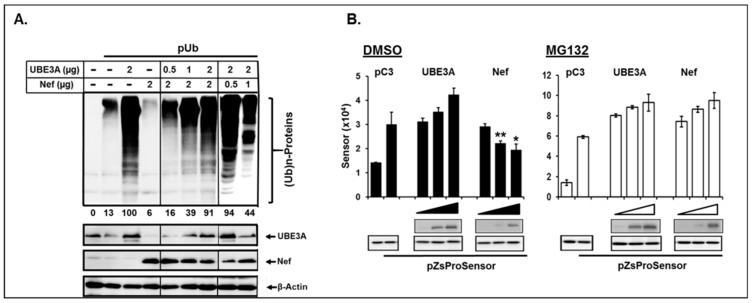
Role of UBE3A and Nef in ubiquitination and proteasome activity. (**A**) Top panel represents ubiquitinated cellular proteins, when the indicated amount of pNef and/or pUBE3 together with 2 μg pUb.HA was transfected, Arabic numbers below the panel indicate relative amount of ubiquitinated cellular protein in each lane, and lower panel shows expression of each protein by WB analysis. (**B**) The sensor activity was measured after 48 h post-transfection with 2 μg isotype plasmid (pCDNA3, pC3) or increasing amount (1, 2, 4 μg) of pUBE3A or pNef together with pZsProSensor, followed by treatment of the cells with 2 μM MG132 (open bar) or its vehicle, DMSO (closed bar). WB analysis under the bar graph represented the amount of the proteins from the cells treated with DMSO and MG132, and all bands were derived from the identical membrane.

**Figure 7 viruses-11-01098-f007:**
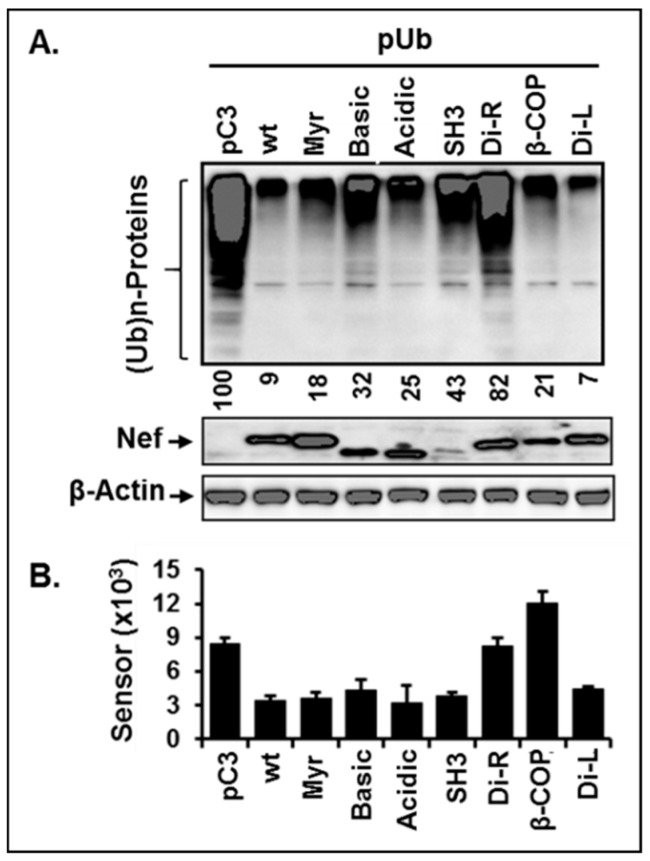
Identification of motif(s) in Nef in regulation of ubiquitination and proteasome activity. Identical amount of the indicated mutant nef plasmid together with pUb was transfected, and the ubiquitinated cellular proteins were analyzed by WB analysis. Arabic number under the image indicated relative amount of the ubiquitinated cellular proteins. The bar graph depicted alteration of the proteasome activity measured as described above by the introduced mutations.

**Figure 8 viruses-11-01098-f008:**
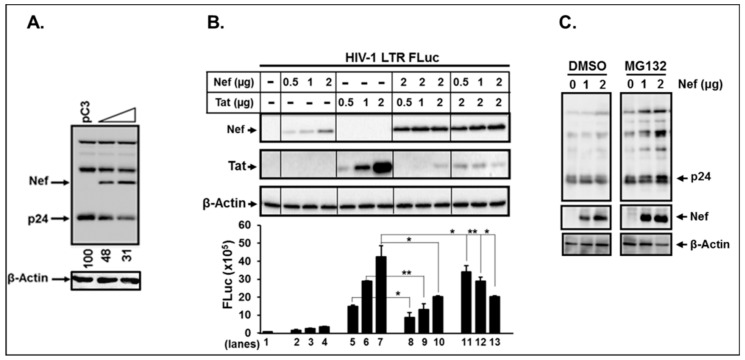
Effects of ectopic expression of Nef on HIV-1 replication and the stabilities of Tat and Gag. Panel (**A**) represented WB analyses with the equivalent amount of cell lysates generated from cells transfected with 0, 1, and 3 µg pNef.Myc and 2 µg of wt- HIV-1, and the Arabic numbers show relative amount of p24. pC3 indicates isotype plasmid, pCDNA3, of pNef.Myc. (**B**) Changes in the amount of Tat in the presence of Nef in the cells transfected with the indicated amount of pTat.Myc and/or pNef.Myc were analyzed by WB analysis (upper), and the impact of Nef on Tat-Triggered HIV-1 LTR promoter activity was monitored by measuring Firefly luciferase activity. This luciferase assay is representative of three independent experiments with triplicates. (**C**) 293T cells were transfected with 2 µg of psPax2 and the indicated amount of pNef.Myc (Nef), the transfected cells were treated with DMSO and MG132, as indicated above, and WB was performed to visualize the indicated proteins. All bands were derived from the identical membrane with the equivalent amount of protein loading.

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
