# Peer review of "HIV-1 Impairment via UBE3A and HIV-1 Nef Interactions Utilizing the Ubiquitin Proteasome System"

_viruses, 2019, doi:10.3390/v11121098_

Round 1

Reviewer 1 Report

Overall Summary: In this manuscript, the authors examine the potential interactions between HIV-1 Nef and the host ubiquitin (Ub)-protein ligase E3A. The authors suggest that this interaction is “is integral to regulation of viral and cellular protein decay and thereby the competition between HIV-1 and host cell survival”.  The manuscript also demonstrates that this interaction not only leads to the degradation of Nef but also to the degradation of other important HIV-1 proteins, including Tat.  This has suggested a role for this complex in coordinating HIV-1 protein degradation during the HIV-1 lifecycle thereby effecting HIV-1 replication in the host cell. While this research is interesting and novel, there are a few areas that need to be addressed prior to publication, most importantly the enhanced discussion of the translational nature of this work and why it is important.

Comments for the authors:

The authors need to expand the methods section to include more pertinent information, and also include necessary information such as number of repeats and replicates in each assay. Figure 2: The authors should also include endogenous and transfected labels on Figure 2A. Also, it does not appear to be 65% reduction.  Possibly enlarge the figure to make it easier to see this change, there would seem to be room to do this. Page 6, lines 217-223: need to make reference to Figure 2C somewhere in this section. Page6, lines 232-236: The authors state that “…increasing UBE3A gradually lowered the amount of RT in the culture supernatants from wt-, but not from Δnef-HIV-1 (data not shown)…”.This reviewer thinks this data is important, lends itself well to the paper, and as such the data should be included as a figure.  Figure 4 legend: The legend for figure 4A does not appear to be referencing the figure as shown. This point should be corrected. Page 14, lines 544-546: The authors mention the potential utility and translation of this research.However, only one sentence is spent discussing its importance. This reviewer would like to see this aspect developed more and further discussed.  The translation of this research is important, so spend some time expanding upon on this aspect of the manuscript would seem worthwhile.  The authors do not seem to provide a clear understanding of exactly how they plan to expand upon this research to begin to effect patient health (therapeutics as they mention) or the further understanding of pathogenesis in patients.  The lack of this appreciably decreases the significance of this work.

Author Response

Reviewer 1.

The authors need to expand the methods section to include more pertinent information, and also include necessary information such as number of repeats and replicates in each assay. Figure 2: The authors should also include endogenous and transfected labels on Figure 2A. Also, it does not appear to be 65% reduction.  Possibly enlarge the figure to make it easier to see this change, there would seem to be room to do this. Page 6, lines 217-223: need to make reference to Figure 2C somewhere in this section. Page6, lines 232-236: The authors state that “…increasing UBE3A gradually lowered the amount of RT in the culture supernatants from wt-, but not from Δnef-HIV-1 (data not shown)…”.This reviewer thinks this data is important, lends itself well to the paper, and as such the data should be included as a figure.  Figure 4 legend: The legend for figure 4A does not appear to be referencing the figure as shown. This point should be corrected. Page 14, lines 544-546: The authors mention the potential utility and translation of this research.However, only one sentence is spent discussing its importance. This reviewer would like to see this aspect developed more and further discussed.  The translation of this research is important, so spend some time expanding upon on this aspect of the manuscript would seem worthwhile.  The authors do not seem to provide a clear understanding of exactly how they plan to expand upon this research to begin to effect patient health (therapeutics as they mention) or the further understanding of pathogenesis in patients.  The lack of this appreciably decreases the significance of this work. 

Submission Date

21 September 2019

Date of this review

11 Oct 2019 17:42:51

We appreciate the reviewer’s points, especially on the translational significance of this research, which we lightly dealt with.  In response to these criticisms, we amended the manuscript as follows, and we hope these changes suffice to answer the reviewer.

To the reviewer’s criticisms, we responded in red as described below,

To the number of repetitions of each assay,

[Response] We've now added the following sentence to Section 2.7 of Materials and Methods, “The IP and/or WB analyses in the figures are representative of multiple independent experiments.”  We also added the following sentence, “This luciferase assay is representative of three independent experiments with triplicates.”, to the bar graphs for the luciferase assays.

To the reviewer’s request on Fig. 2A,

[Response] We added the “endogenous” and “transfected” labels to the Figure 2A and explained the 65 % reduction in the corresponding figure legend by adding the following sentence: “The amount of UBE3A shown as an Arabic number under the Figure was standardized with the changes of amount of β-Actin.”  To elaborate, the Bio-Rad image tool for quantification shows that the band intensity of UBE3A with (+) siRNA/3A was approximately 73 % of that with (-) siRNA/3A, even if the total amount of proteins loaded on the gel, based on the amount of β-Actin, were 207 % (2.07 fold) more than those in the absence of siRNA/3A, yielding an actual UBE3A amount in (+) of 35.2 %, compared with (-).

To make reference to Figure 2C,

[Response] We've now added (Fig. 2C) in line 226.

To the addition of the figure reflecting the amount of RT,

[Response] We agree with the reviewer’s opinion, and thus we added the RT data in the same figure (Figure 4) and changed the text, accordingly.

In discussion of the translational significance of this research,

[Response] We have thus added the following paragraph to raise the translational significance of our research to the end of Discussion.  “Specifically, since changes in the intact UBE3A protein expression could dysregulate UBE3A-associated proteasomal degradation and thus HIV-infected cell biology, a modified UBE3A, composed strictly of the Nef binding domain and the active motif for specific degradation of viral proteins, could be employed to promote UBE3A-triggered targeted knockout of HIV-1 proteins, and to impact the virus/cell competition for survival in the host”. 

Reviewer 2 Report

In this paper, Pyeon et al show that UBE3A suppresses HIV 1 replication by reducing the levels of p24 in infected cells. Expression of UBE3A reduces the replication of HIV, detected by p24 accumulation as well as virus production, but not of an HIV-∆nef, conclusively showing that the UBE3A repression of HIV replication is nef-dependent. While The authors provide convincing evidence for the induction of the proteasome-mediated degradation of nef by UBE3A, the reverse ie induction of proteasome mediated degradation of UBE3A by nef expression are not convincing, as discussed in the major points. The authors next show that nef reduced cellular proteins ubiquitination, and enhance proteasome function, while UBE31 enhanced cellular proteins ubiquitination without altering the function of the proteasome. Distinct domains of nef are shown to be involved in inhibition of ubiquitination and stimulation of proteasome activity. Last, it is shown that nef-mediated reduction of p24 is the result of nef mediated degradation of TAT, and subsequent reduction of viral genes expression.

The work includes interesting results, which are lacking important controls in some cases. In addition, the text and figure legends are lacking clarity in several instances as detailed below.

 Major points

 - figure 1-C primary immune-precipitated proteins have to be presented, please show the IP Myc WB Myc and IP Flag WB Flag. It is not possible to use UBE3A Ab for IP? Owing to the levels of endogenous UBE3A detected, is the endogenous immuno-precipitation possible?

- figure 2-B control with NT siRNA is required to conclude that siUBE3A increases p24 levels

- line 197-198 To affirm that demonstrating that “the observed blockade of HIV-1 replication was UBE3A specific”, transfection of an siRNA-resistant UBE3A expression plasmid in siRNA-transfected cells has to be done, and reverse the effect of siRNA UBE3A.  

- figure 3-A improve legend:  show the amount of transfected UBE3A above the panel (as in fig.4-A) pC3 = non-infected non-transfected: show the UBE3A WB, which will give endogenous level of the protein

- the reduction of UBE3A observed upon ectopic expression of nef in figure 4-A is far lower that observed in cells infected with nef-expressing HIV, as compared to nef-deleted HIV, observed in figure 3-A. Authors should first definitively give the amount of transfected plasmids in fig. 3-A to provide clear comparison between these two figures, they also should discuss this point. In infected cells, other proteins could be involved in the decrease of UBE3A levels.

- line 281: The effect on the LTR-promoter activity could also be due to competitive binding of UBE3A and nef, since both factors regulate HIV LTR. Discuss it.

- figure 5-A: in contrast to author’s claim, a decrease of UBE3A levels in nef co-expressing cells is not observed in the presented results. The current data do not support the nef-mediated reduction since they are only deduced from the relief of an abnormal increase of UBE3A levels upon CHX treatment, whereas a constant level of UBE3A is observed upon co-expression, which would reflect a stable UBE3A protein.

- line 375: “wt-nef enhanced proteasome activity, reducing cleavage of the sensor molecule” should be “increasing cleavage…”

- Figure 5.B, lines 318-319: The data for UBE3A-mediated degradation of nef are convincing, since upon MG132 treatment UBE3A expression do not impact any more the levels of nef. In contrast, it can’t be concluded that nef reduced UBE3A levels through proteasome, since upon MG132 treatment the levels of UBE3A are still greatly decreased upon nef expression. MG132 strongly increases UBE3A levels, which indicates that UBE3A stability is regulated by the proteasome. However, I’m not convinced that the nef protein regulates such proteasome-induced UBE3A degradation.

- Figure 6-A: show the effect of increasing amount of UBE3A expression vector, 0.5, 1, and 2 µg without nef expression to be able to conclude that the nef-mediated inhibition of proteins ubiquitination is alleviated by increasing levels of UBE3

- lines 378-382: above all the different expression levels could results for different accumulation levels, owing to different stabilities of the nef mutated proteins. Simplify this part and state that point.

- minor

- line 51: the degradation of Gag by TRIM5 is not a viruse’s exploitation of cell machinery but rather a defense mechanism of the cell

- line 68-69: sentence not clear, reword please

- line 160 CAT activity is not strong!

- line 167 fig 1-C instead of figure 1-D

- line 203 remove “the infected”

- line 222:  the activation of LTR promoter activity by UBE3A is not slight, but even stronger than that of nef

- line 227-229 shorten the sentence, make 2≠

- line 233: ”amount of RT in the supernatant” means “amount of virus particles in the supernatant determined by RT assays? Please edit.

- figure 1-A it is not clear if the controls with PalT have been done with fusions with B42-AD

- line 237-238 it is not clear at this point why the ENV is included with Gag as target to be degraded by UBE3A. Should have been used as a control, then reword the sentence

- lines 244-245 remove the conclusive sentence which does not fit here

- lines 252-253 “The above data demonstrated that ectopic expression of UBE3A together with Nef restricted HIV-1 replication…”  remove this part of the sentence

- lines 260-262: remove first part of the sentence as follows “These data suggest that the level of UBE3A and Nef is critical to modulate HIV-1 replication in HIV-1-infected cells and that the amount of the intracellular UBE3A was regulated by Nef.

- figure 4-A the amount of plasmids transfected indicated in the figure legend (1, 2, 4 µg)  and in the panel (0,5, 1, 2 µg)  do not fit. Symbols of quantifications presented in panel B not adequately listed in the legend (there are no closed triangles!)

- lines 269-271: sentence not clear, reword

- line 305 refer to -Fig. 5A instead of Fig.5B

- lines 324-325: reword the description of nef regarding ubcH7, which is not understandable. In addition, does not seem that the cited paper (reference 15) is correct;

- line “328-329: “evaluating the capacity of Nef and UBE3A for ubiquitination of target proteins” does not mean anything

- lines 331-332 and Figure 6-B The relief of UBE3A induction of ubiquitination by nef is precisely not dose dependent but appears more at 1 µg than at 2 µg of nef expression plasmid! Remove the statement

- lines 349-350: remove “similar to the above data”

- line 351-362: the hypothesis is that nef reduces the levels of cellular proteins ubiquitination by enhancing their proteasome-mediated degradation? If so, replace “regulating” by “enhancing” in the conclusion sentence.

- lines 429-430: the same experiment in the presence of MG132 could help to reinforce this conclusion

- lines 449- 450: the effect of MG132 on nef and gag do not account for the drawn conclusion, remove it from the sentence.

- lines 494-495: sentence not correct. In Fig. cells transfected with PC3 have the level of endogenous UBE3A but not below (siRNA transfected cells do)

- lines 512-513: remove “previous research showed that”

- lines 519-520: not clear to me what authors mean

Author Response

Top of Form

In this paper, Pyeon et al show that UBE3A suppresses HIV 1 replication by reducing the levels of p24 in infected cells. Expression of UBE3A reduces the replication of HIV, detected by p24 accumulation as well as virus production, but not of an HIV-∆nef, conclusively showing that the UBE3A repression of HIV replication is nef-dependent. While The authors provide convincing evidence for the induction of the proteasome-mediated degradation of nef by UBE3A, the reverse ie induction of proteasome mediated degradation of UBE3A by nef expression are not convincing, as discussed in the major points. The authors next show that nef reduced cellular proteins ubiquitination, and enhance proteasome function, while UBE31 enhanced cellular proteins ubiquitination without altering the function of the proteasome. Distinct domains of nef are shown to be involved in inhibition of ubiquitination and stimulation of proteasome activity. Last, it is shown that nef-mediated reduction of p24 is the result of nef mediated degradation of TAT, and subsequent reduction of viral genes expression.

The work includes interesting results, which are lacking important controls in some cases. In addition, the text and figure legends are lacking clarity in several instances as detailed below.

[Response]  We sincerely appreciate the reviewer’s careful criticisms of our submission, which point to scientific and presentational oversights that we needed to address. We hence revised our manuscript and believe it now to be sufficiently comprehensive and convincing. 

 Major points

- figure 1-C primary immune-precipitated proteins have to be presented, please show the IP Myc WB Myc and IP Flag WB Flag. It is not possible to use UBE3A Ab for IP? Owing to the levels of endogenous UBE3A detected, is the endogenous immuno-precipitation possible?

[Response]  In response to the reviewer’s request, the second panel of Fig. 1C (WB: Myc) is now replaced with IP:Myc/WB: Myc.  However, anti-UBE3A could not efficiently precipitate UBE3A, and thus we employed anti-Flag for precipitation of the protein.

 - figure 2-B control with NT siRNA is required to conclude that siUBE3A increases p24 levels

[Response]  Our independent experiment shows that the scrambled siRNA (s-siRNA) purchased from Sigma as a control for siRNA/3A was employed to test the specificity of siRNA/3A.  Our data show that the amount of UBE3A was basically unchanged, as shown in the insert, wherein the bands indicate the expressed UBE3A in the absence of s-siRNA (left lane) and  the increasing amount of s-siRNA corresponding to the amount of siRNA/3A in the Figure (next 2 lanes).  These data evince the specificity of siRNA/3A, and thus we added a phrase in the 192nd line, “but not by its scrambled control siRNA/3A (data not shown),”.                                                                                     

 - line 197-198 To affirm that demonstrating that “the observed blockade of HIV-1 replication was UBE3A specific”, transfection of an siRNA-resistant UBE3A expression plasmid in siRNA-transfected cells has to be done, and reverse the effect of siRNA UBE3A.

[Response] We wonder what is the advantage of siRNA expressed from the transfected plasmid vs our commercial siRNA whose specificity, not activity, has already been tested. 

 - figure 3-A improve legend:  show the amount of transfected UBE3A above the panel (as in fig.4-A) pC3 = non-infected non-transfected: show the UBE3A WB, which will give endogenous level of the protein

[Response]  As the reviewer suggests, the figure is amended by adding the amount of UBE3A on the top and at the bottom of the figure and the following sentence, “The total amount of the transfected DNA was adjusted with the isotype plasmid pC3 level remaining the same.” to the Figure.

- the reduction of UBE3A observed upon ectopic expression of nef in figure 4-A is far lower that observed in cells infected with nef-expressing HIV, as compared to nef-deleted HIV, observed in figure 3-A. Authors should first definitively give the amount of transfected plasmids in fig. 3-A to provide clear comparison between these two figures, they also should discuss this point. In infected cells, other proteins could be involved in the decrease of UBE3A levels.

[Response] According to our data, the absolute amount of intracellular UBE3A is changed by the amount of Nef, and the amount of Nef in Fig. 4A is not directly comparable with that of Nef in 3A since that of Fig. 3A is expressed from HIV-1-replicating 293T, while Nef in Fig. 4A is expressed from the transfected nef-expressing plasmid.  Thus, the intensity of UBE3A bands in Figs. 3A and 4A is expected to differ.  We briefly mention this point at the corresponding place in the “Results” section: “However, the reductions in band intensities of UBE3A in Figs. 3A and 4A were disparate, which could be due to the differential amount of Nef expressed from the wt-HIV-1 replication and from nef-expressing plasmid, respectively.”

- line 281: The effect on the LTR-promoter activity could also be due to competitive binding of UBE3A and nef, since both factors regulate HIV LTR. Discuss it.

[Response] There could be alternative explanations to describe the observed phenomenon.  However, competitive binding of UBE3A and Nef -- unless we provide that both proteins bind to the identical cis-acting element in the LTR -- is beyond the scope of interpretation/discussion with the current data.

- figure 5-A: in contrast to author’s claim, a decrease of UBE3A levels in nef co-expressing cells is not observed in the presented results. The current data do not support the nef-mediated reduction since they are only deduced from the relief of an abnormal increase of UBE3A levels upon CHX treatment, whereas a constant level of UBE3A is observed upon co-expression, which would reflect a stable UBE3A protein.

[Response]  Fig. 5A clearly indicates that the amount of the synthesized UBE3A and Nef was not reduced in the presence of CHX, when they were expressed independently (Fig. 5A, top, left panel), whereas both of the synthesized UBE3A and Nef were reduced as the treatment time was extended, when they were co-expressed (Fig. 5A, top, right panel).

- line 375: “wt-nef enhanced proteasome activity, reducing cleavage of the sensor molecule” should be “increasing cleavage…”

[Response] We replaced “reducing” with “increasing”.

 - Figure 5.B, lines 318-319: The data for UBE3A-mediated degradation of nef are convincing, since upon MG132 treatment UBE3A expression do not impact any more the levels of nef. In contrast, it can’t be concluded that nef reduced UBE3A levels through proteasome, since upon MG132 treatment the levels of UBE3A are still greatly decreased upon nef expression. MG132 strongly increases UBE3A levels, which indicates that UBE3A stability is regulated by the proteasome. However, I’m not convinced that the nef protein regulates such proteasome-induced UBE3A degradation.

[Response] The levels of both UBE3A and Nef are significantly increased, when the cells were treated with MG132, indicating that accumulation of these proteins was due to the blockage of the proteins from proteasomal degradation.

-Figure 6-A: show the effect of increasing amount of UBE3A expression vector, 0.5, 1, and 2 µg without nef expression to be able to conclude that the nef-mediated inhibition of proteins ubiquitination is alleviated by increasing levels of UBE3

 [Response]  It is very well established that E3 ligase, including UBE3A, enhances ubiquitination of the target proteins.  Consistently, our data in Fig. 6A show that UBE3A dramatically augmented ubiquitination of the cellular protein (the 3rd lane of the blot from the left), even if we did not use different doses of UBE3A,    

- lines 378-382: above all the different expression levels could results for different accumulation levels, owing to different stabilities of the nef mutated proteins. Simplify this part and state that point.

[Response] What we mean is that the different amount of Nef and its mutants in the Figure could be due to the differential extraction efficiency of the expressed mutant nef proteins by the lysis buffer, differential subcellular localization of the mutant proteins, and/or the affinity changes of the mutant Nef to the antibody.  However, we do not know at this point that introduction of the indicated mutation in Nef affects stability of the mutant proteins, which could be a good subject to study in ensuing experiments.  Thus, we amended the sentence slightly, to be more comprehensive.

 - minor

 - line 51: the degradation of Gag by TRIM5 is not a viruse’s exploitation of cell machinery but rather a defense mechanism of the cell

[Response] In the sentence, we did not address “virus exploitation” but “HIV-1 and HIV-1-infected cells exploit~~” .  Further, we state that “degradation of Gag by TRIM5a is integral to species tropism”, not that  “virus exploits cell machinery for its defense.”

- line 68-69: sentence not clear, reword please

[Response] We would appreciate the reviewer to more specifically point out the unclear part.

- line 160 CAT activity is not strong!

[Response]  Part of the data (the yeast and mammalian 2- hybrid) in Fig. 1 was generated a long time ago -- and adequately to the standards of that day -- when the first and corresponding author worked at Harvard Medical School, and before the particular CAT assay system and visual outputs then commercially available were fully replaced by the higher performance tools/reagents and improved optical readouts that our discipline enjoys today. We therefore have now revisited the assay's visual output for the discussed figure, with modern parameters for a more sensitive readout, which we find confirmatory in terms of signal relative to background.

- line 167 fig 1-C instead of figure 1-D

[Response] We changed 1D to 1C.

- line 203 remove “the infected”

[Response] We removed “the infected” from the sentence, “~thereby suppressing replication of the infected HIV-1 ~”.

- line 222:  the activation of LTR promoter activity by UBE3A is not slight, but even stronger than that of nef

[Response] We agree with the reviewer’s point that UBE3A-mediated LTR promoter activity was higher than Nef.  However, the significance of the differential promoter activity by UBE3A from by Nef is unclear at this point.  What we emphasize here is suggesting that the observed inhibition of HIV-1 replication by UBE3A was not due to the impairment of viral gene expression.

- line 227-229 shorten the sentence, make 2≠

[Response]  We divided the sentence in two to avoid the over-complication.

- line 233: ”amount of RT in the supernatant” means “amount of virus particles in the supernatant determined by RT assays? Please edit.

[Response] We amended the sentence, as the reviewer suggests.

- figure 1-A it is not clear if the controls with PalT have been done with fusions with B42-AD

[Response] Yes, it is fused with B42AD.

- line 237-238 it is not clear at this point why the ENV is included with Gag as target to be degraded by UBE3A. Should have been used as a control, then reword the sentence

[Response] To make the sentence clearer, we added a phrase, “as another structural protein control”, in between Env and can in the 241st line of the revised manuscript.

- lines 244-245 remove the conclusive sentence which does not fit here

[Response]  The sentence is now removed.

- lines 252-253 “The above data demonstrated that ectopic expression of UBE3A together with Nef restricted HIV-1 replication…”  remove this part of the sentence.

[Response]  The sentence is removed from the indicated sentence.

- lines 260-262: remove first part of the sentence as follows “These data suggest that the level of UBE3A and Nef is critical to modulate HIV-1 replication in HIV-1-infected cells and that the amount of the intracellular UBE3A was regulated by Nef.

[Response]  The sentence is removed from the 263rd sentence.

- figure 4-A the amount of plasmids transfected indicated in the figure legend (1, 2, 4 µg)  and in the panel (0,5, 1, 2 µg)  do not fit. Symbols of quantifications presented in panel B not adequately listed in the legend (there are no closed triangles!)

[Response]  We appreciate this reviewer’s points on our lapse in the figure.  We have now corrected this.

- lines 269-271: sentence not clear, reword

[Response]  The sentence is rephrased.

- line 305 refer to -Fig. 5A instead of Fig.5B

[Response] Fig. 5B is substituted with Fig. 5A.

- lines 324-325: reword the description of nef regarding ubcH7, which is not understandable. In addition, does not seem that the cited paper (reference 15) is correct;

[Response] We appreciate the reviewer’s observation on the reference and now correct the citation.

- line “328-329: “evaluating the capacity of Nef and UBE3A for ubiquitination of target proteins” does not mean anything

[Response] We reworded the sentence. 

- lines 331-332 and Figure 6-B The relief of UBE3A induction of ubiquitination by nef is precisely not dose dependent but appears more at 1 µg than at 2 µg of nef expression plasmid! Remove the statement

[Response] We removed the “dose-dependent fashion”.  We also changed “three” to “two” in the 336th line.

- lines 349-350: remove “similar to the above data”

[Response]  We removed the phrase.

- line 351-362: the hypothesis is that nef reduces the levels of cellular proteins ubiquitination by enhancing their proteasome-mediated degradation? If so, replace “regulating” by “enhancing” in the conclusion sentence.

[Response]  The word is now changed, as the reviewer suggested.

- lines 429-430: the same experiment in the presence of MG132 could help to reinforce this conclusion

[Response]  We added the following sentence, “Further experiments are needed to clarify the mechanism of Nef-mediated proteasomal degradation of Tat, which will raise our understanding of the complex molecular actions of Nef.” at the end of the line 527 in Discussion.

- lines 449- 450: the effect of MG132 on nef and gag do not account for the drawn conclusion, remove it from the sentence.

[Response] We remove the phrase.

- lines 494-495: sentence not correct. In Fig. cells transfected with PC3 have the level of endogenous UBE3A but not below (siRNA transfected cells do)

[Response]  The pC3 was substituted with (Fig. 2B, Mock) to make the sentence clear.

- lines 512-513: remove “previous research showed that”

[Response] We removed “Previous research showed” from the sentence.

- lines 519-520: not clear to me what authors mean

[Response] The sentence is rephrased to be more comprehensive.

Submission Date

21 September 2019

Date of this review

06 Oct 2019 20:54:41

Bottom of Form

Round 2

Reviewer 2 Report

Overall the manuscript is well improved. Please find below replies to several  author's answers

figure 2-B control with NT siRNA is required to conclude that siUBE3A increases p24 levels

[Response]  Our independent experiment shows that the scrambled siRNA (s-siRNA) purchased from Sigma as a control for siRNA/3A was employed to test the specificity of siRNA/3A.  Our data show that the amount of UBE3A was basically unchanged, as shown in the insert, wherein the bands indicate the expressed UBE3A in the absence of s-siRNA (left lane) and  the increasing amount of s-siRNA corresponding to the amount of siRNA/3A in the Figure (next 2 lanes).  These data evince the specificity of siRNA/3A, and thus we added a phrase in the 192nd line, “but not by its scrambled control siRNA/3A (data not shown),”.       

[reviewer reply] Here the point of the reviewer was not to ascertain the specificity of the siRNA, but to   provide a strict comparison between the level of p24 in cells that have been treated in parallel with the siRNA/3A and non-target siRNA, meaning to perform the experiment as described in the text : “we infected HIV-1 into Jurkat, and the infected cells were expanded  by 3-fold dilution with the fresh RPMI1640 at every 3 day, while measuring RT activity in the culture supernatants. The infected cells were then equally aliquoted on 6 days post-infection, and the aliquoted cells were transfected with different doses of pUBE3A, of siRNA/3A or of non-target siRNA”. This would provide a direct comparison of the effect of UBE3A silencing upon treatment with siRNA (ie the control corresponding to pC3 transfection for UBE3A expression plasmids transfection).

 - line 197-198 To affirm that demonstrating that “the observed blockade of HIV-1 replication was UBE3A specific”, transfection of an siRNA-resistant UBE3A expression plasmid in siRNA-transfected cells has to be done, and reverse the effect of siRNA UBE3A.

[Response] We wonder what is the advantage of siRNA expressed from the transfected plasmid vs our commercial siRNA whose specificity, not activity, has already been tested. 

[reviewer reply] Because the infected cells have been either transfected with pUBE3A or with siRNA/3A, it is hard to conclude that “UBE3A siRNA treatment expunged UBE3A-triggered inhibitory effects on p24 accumulation”, which was used by the authors as a pointdemonstrating that the observed blockade of HIV-1 replication was UBE3A-specific” To strictly demonstrate such specificity, as proposed in the previous review, authors should transfect an expression plasmids for UBE3A that is not silenced by siRNA/3A and show that the siRNA do not any more reduce p24 level. That being said, the reviewer would only suggest to dampen the conclusion in the text and replace “demonstrating” by “suggesting”  

[Response] What we mean is that the different amount of Nef and its mutants in the Figure could be due to the differential extraction efficiency of the expressed mutant nef proteins by the lysis buffer, differential subcellular localization of the mutant proteins, and/or the affinity changes of the mutant Nef to the antibody.  However, we do not know at this point that introduction of the indicated mutation in Nef affects stability of the mutant proteins, which could be a good subject to study in ensuing experiments.  Thus, we amended the sentence slightly, to be more comprehensive.

[reply] I understood well what was ment by the authors, but still think that the first hypothesis is that the mutated nef proteins have different accumulation levels.

Author Response

Overall the manuscript is well improved. Please find below replies to several  author's answers

figure 2-B control with NT siRNA is required to conclude that siUBE3A increases p24 levels

[Response]  Our independent experiment shows that the scrambled siRNA (s-siRNA) purchased from Sigma as a control for siRNA/3A was employed to test the specificity of siRNA/3A.  Our data show that the amount of UBE3A was basically unchanged, as shown in the insert, wherein the bands indicate the expressed UBE3A in the absence of s-siRNA (left lane) and  the increasing amount of s-siRNA corresponding to the amount of siRNA/3A in the Figure (next 2 lanes).  These data evince the specificity of siRNA/3A, and thus we added a phrase in the 192nd line, “but not by its scrambled control siRNA/3A (data not shown),”.       

[reviewer reply] Here the point of the reviewer was not to ascertain the specificity of the siRNA, but to   provide a strict comparison between the level of p24 in cells that have been treated in parallel with the siRNA/3A and non-target siRNA, meaning to perform the experiment as described in the text : “we infected HIV-1 into Jurkat, and the infected cells were expanded  by 3-fold dilution with the fresh RPMI1640 at every 3 day, while measuring RT activity in the culture supernatants. The infected cells were then equally aliquoted on 6 days post-infection, and the aliquoted cells were transfected with different doses of pUBE3A, of siRNA/3A or of non-target siRNA”. This would provide a direct comparison of the effect of UBE3A silencing upon treatment with siRNA (ie the control corresponding to pC3 transfection for UBE3A expression plasmids transfection).

[Response] We amended the text, as the reviewer requested.

 - line 197-198 To affirm that demonstrating that “the observed blockade of HIV-1 replication was UBE3A specific”, transfection of an siRNA-resistant UBE3A expression plasmid in siRNA-transfected cells has to be done, and reverse the effect of siRNA UBE3A.

[Response] We wonder what is the advantage of siRNA expressed from the transfected plasmid vs our commercial siRNA whose specificity, not activity, has already been tested. 

[reviewer reply] Because the infected cells have been either transfected with pUBE3A or with siRNA/3A, it is hard to conclude that “UBE3A siRNA treatment expunged UBE3A-triggered inhibitory effects on p24 accumulation”, which was used by the authors as a point “demonstrating that the observed blockade of HIV-1 replication was UBE3A-specific” To strictly demonstrate such specificity, as proposed in the previous review, authors should transfect an expression plasmids for UBE3A that is not silenced by siRNA/3A and show that the siRNA do not any more reduce p24 level. That being said, the reviewer would only suggest to dampen the conclusion in the text and replace “demonstrating” by “suggesting”  

[Response]  As the reviewer suggested, we replaced “demonstrating” with “suggesting”.

[Response] What we mean is that the different amount of Nef and its mutants in the Figure could be due to the differential extraction efficiency of the expressed mutant nef proteins by the lysis buffer, differential subcellular localization of the mutant proteins, and/or the affinity changes of the mutant Nef to the antibody.  However, we do not know at this point that introduction of the indicated mutation in Nef affects stability of the mutant proteins, which could be a good subject to study in ensuing experiments.  Thus, we amended the sentence slightly, to be more comprehensive.

[reply] I understood well what was ment by the authors, but still think that the first hypothesis is that the mutated nef proteins have different accumulation levels.

[Response] We fully understand the reviewer’s concern and thus repeated this experiment at least 3-4 times after re-quantification of the mutant nef-expressing plasmids by UV spectrophotometer followed by agarose gel electrophoresis.  We consistently obtained identical pattern of WB each time.  However, as the reviewer pointed out, since we cannot exclude the possibility of accumulation level of the mutant Nef, we added a phrase, “and/or different level of accumulation of the mutant Nefs” in the sentence.

Round 3

Reviewer 2 Report

The paper has been adequately edited